# Quantum revivals in HgTe/CdTe quantum wells and topological phase transitions

Alberto Mayorgas*

*Department of Applied Mathematics, University of Granada, Fuentenueva s/n, 18071 Granada, Spain*

Manuel Calixto

*Department of Applied Mathematics, University of Granada, Fuentenueva s/n, 18071 Granada, Spain and*
*Institute Carlos I for Theoretical for Theoretical and Computational Physics (iC1), Fuentenueva s/n, 18071 Granada, Spain*

Nicolás A. Cordero

*Department of Physics, University of Burgos, 09001 Burgos, Spain*
*International Research Center in Critical Raw Materials for Advanced Industrial*
*Technologies (ICCRAM), University of Burgos, 09001 Burgos, Spain and*
*Institute Carlos I for Theoretical for Theoretical and Computational Physics (iC1), Fuentenueva s/n, 18071 Granada, Spain*

Elvira Romera

*Department of Atomic, Molecular and Nuclear Physics,*
*University of Granada, Fuentenueva s/n, 18071 Granada, Spain and*
*Institute Carlos I for Theoretical for Theoretical and Computational Physics (iC1), Fuentenueva s/n, 18071 Granada, Spain*

Octavio Castaños

*Institute of Nuclear Sciences, National Autonomous University of Mexico, Apdo. Postal 70-543, 04510, CDMX, Mexico*
(Dated: April 19, 2024)

The time evolution of a wave packet is a tool to detect topological phase transitions in two-dimensional Dirac materials, such as graphene and silicene. Here we extend the analysis to HgTe/CdTe quantum wells and study the evolution of their electron current wave packet, using 2D effective Dirac Hamiltonians and different layer thicknesses. We show that the two different periodicities that appear in this temporal evolution reach a minimum near the critical thickness, where the system goes from normal to inverted regime. Moreover, the maximum of the electron current amplitude changes with the layer thickness, identifying that current maxima reach their higher value at the critical thickness. Thus, we can characterize the topological phase transitions in terms of the periodicity and amplitude of the electron currents.

## I. INTRODUCTION

The time evolution of wave packets can have interesting behaviors due to quantum interference. Revivals occur when a well-localized wave-packet evolves in time to recover, at least approximately, its initial waveform. This event occurs periodically and the period is known as the revival period. The phenomenon of quantum wave packet revivals has been investigated theoretically in atomic systems, molecules, many body systems or 2D Materials [1–11] and observed experimentally in among others, Rydberg atoms or molecular systems [12–16]. Recently, it has been shown how revival and classical periods reveal quantum phase transitions in many-body systems [5, 6]. Furthermore, it has also been seen how both periods are capable of detecting topological phase transitions (TPTs for short) in two-dimensional materials such as graphene [4] and silicene [7].

In this work, we focus on a particular zincblende heterostructure, the mercury telluride-cadmium telluride (HgTe/CdTe) quantum wells (QWs). They have been widely used to study the quantum spin Hall effect and

new types of topological phases [17–20], and traditionally are part of optical and transport experiments involving spin-related observations [21–23]. At present, HgTe/CdTe QWs appear together with other topological insulators to construct low-dimensional quantum devices, which can experimentally realize quantum anomalous Hall effects [24–28]. One of the most interesting properties of these materials is that we can switch between normal or inverted band structures by simply changing the QW width (layer thickness in our jargon). In particular, we study the time evolution of electron current wave packets in HgTe/CdTe QWs in magnetic fields, for different values of the HgTe layer thickness to characterize TPTs. We analyze the periodicities in the dynamics of the wave packets and the amplitude of the electron currents. There are other ways to detect topological-band insulator phase transitions, such as information or entropic measures [29–33], or magneto-optical properties [34–39]. In contrast to these methods, quantum revivals provide an straightforward approach to TPTs that has not been applied to HgTe/CdTe QWs so far.

This paper is organized as follows. In the next section we will describe the 2D effective Dirac Hamiltonian for surface states in HgTe/CdTe QWs. In the third section we will study the relation between wave packet revivals

* albmayrey97@ugr.es

| $\lambda$ | $\alpha$ | $\beta$ | $\delta$ | $\mu$ | $\Delta$ |
|---|---|---|---|---|---|
| (nm) | (meV.nm) | (meV.nm$^2$) | (meV.nm$^2$) | (meV) | (meV) |
| 5.5 | 387 | -480 | -306 | 9 | 1.8 |
| 6.1 | 378 | -553 | -378 | -0.15 | 1.7 |
| 7.0 | 365 | -686 | -512 | -10 | 1.6 |

TABLE I. Different values of the HgTe/CdTe QW expansion parameters depending on the HgTe layer thicknesses $\lambda$ [40, 42].

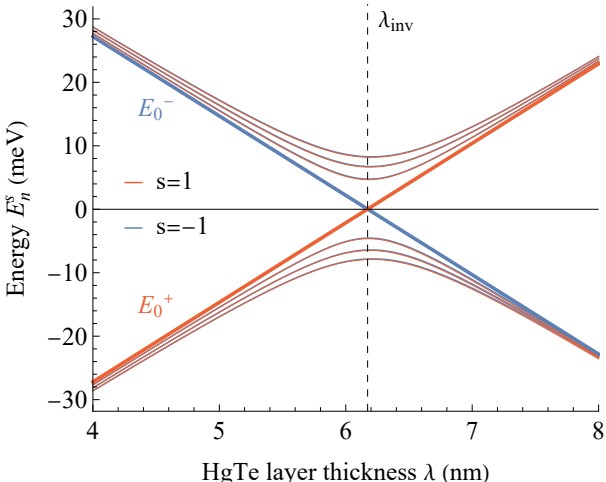

FIG. 1. HgTe/CdTe quantum well low-energy spectrum $E_n^s$ for $B = 0.05$ T, as a function of the HgTe layer thickness $\lambda$. The thin solid lines represent Landau levels $n = \pm 1, \pm 2, \pm 3$ (valence $(-)$ and conduction $(+)$) for spin $s = -1$ (blue) and $s = +1$ (red), and the thick lines represent edge states ($n = 0$). A vertical dashed black line indicates the HgTe thickness $\lambda_{\text{inv}}(0.05) = 6.173$ nm $\simeq \lambda_c$ where the band inversion for edge states occurs for $B = 0.05$ T according to (15).

and classical periodicities with topological phase transition. The relation between the evolution of the electron currents and the TPTs will be described in Section IV. Section V briefly discusses the effect of spin-orbit interaction. The final section is devoted to conclusions.

## II. HgTe/CdTe QUANTUM WELLS LOW-ENERGY HAMILTONIAN

We shall use a 2D effective Dirac Hamiltonian to describe the surface states in HgTe/CdTe QWs, following the prescription of the references [17–20],

$$H = \begin{pmatrix} H_{+1} & 0 \\ 0 & H_{-1} \end{pmatrix}, \ H_s(\boldsymbol{k}) = \epsilon_0(\boldsymbol{k})\tau_0 + \boldsymbol{d}_s(\boldsymbol{k}) \cdot \boldsymbol{\tau}, \quad (1)$$

where $\tau_i$ are the Pauli matrices, $s = \pm 1$ is the spin and $H_{-1}(\boldsymbol{k}) = H_{+1}^*(-\boldsymbol{k})$ (temporarily reversed). It is convenient to expand the Hamiltonian $H_s(\boldsymbol{k})$ around the center $\Gamma$ of the first Brillouin zone [18],

$$\epsilon_0(\boldsymbol{k}) = \gamma - \delta\boldsymbol{k}^2, \quad \boldsymbol{d}_s(\boldsymbol{k}) = (\alpha s k_x, \alpha k_y, \mu - \beta\boldsymbol{k}^2), \quad (2)$$

where $\alpha, \beta, \gamma, \delta$ and $\mu$ are expansion parameters that depend on the HgTe layer thickness $\lambda$, as can be found in [40] and in Table II. Among all these parameters, we highlight the mass or gap term $\mu$ related to the magnetic moment, and the Wilson term $\beta\boldsymbol{k}^2$ (introduced to avoid the Fermion doubling problem [41]). The parameter $\gamma$ can be neglected and we shall take it equals to zero in all calculations.

For $s = \pm 1$, the energy of the valence and conduction bands is

$$\epsilon_\pm(\boldsymbol{k}) = \epsilon_0(\boldsymbol{k}) \pm \sqrt{\alpha^2\boldsymbol{k}^2 + (\mu - \beta\boldsymbol{k}^2)^2}. \quad (3)$$

To differentiate between band insulator and topological insulator phases, one can use the Thouless-Kohmoto-Nightingale-Nijs (TKNN) formula [43] providing the Chern-Pontryagin number $\mathcal{C}$. In the case of the HgTe QWs (see [39] for more details),

$$\mathcal{C}_s = s[\text{sign}(\mu) + \text{sign}(\beta)]. \quad (4)$$

The Chern number depends on the sign of the material parameters $\mu$ and $\beta$, and on the spin $s$. Considering Table II, only $\mu$ changes sign for different layer thicknesses $\lambda$,

thus, the TPT is governed by $\text{sign}(\mu)$, or by $\text{sign}(\mu/\beta)$ as can be found in the literature [40]. Namely, around the critical HgTe layer thickness $\lambda_c \approx 6.1$ nm, the system goes from normal ($\lambda < \lambda_c$ or $\mu/\beta < 0$) to the inverted ($\lambda > \lambda_c$ or $\mu/\beta > 0$) regimes.

We introduce the interaction with a perpendicular magnetic field $B$ along the $z$-axis using minimal coupling $\boldsymbol{p} \rightarrow \boldsymbol{P} = \boldsymbol{p} + e\boldsymbol{A}$, where $\boldsymbol{A} = (A_x, A_y) = (-By, 0)$ is the electromagnetic potential in the Landau gauge, $e$ the electron charge, and $\boldsymbol{p}$ the momentum operator ($\boldsymbol{k} \rightarrow \boldsymbol{p}/\hbar$). Using Peierls' substitution [44, 45], the Hamiltonian (1) is written in terms of creation $a^\dagger$ and annihilation $a$ operators [39],

$$H_{+1} = \begin{pmatrix} \gamma + \mu - \frac{(\delta+\beta)(2N+1)}{\ell_B^2} & \frac{\sqrt{2}\alpha}{\ell_B}a \\ \frac{\sqrt{2}\alpha}{\ell_B}a^\dagger & \gamma - \mu - \frac{(\delta-\beta)(2N+1)}{\ell_B^2} \end{pmatrix},$$

$$H_{-1} = \begin{pmatrix} \gamma + \mu - \frac{(\delta+\beta)(2N+1)}{\ell_B^2} & -\frac{\sqrt{2}\alpha}{\ell_B}a^\dagger \\ -\frac{\sqrt{2}\alpha}{\ell_B}a & \gamma - \mu - \frac{(\delta-\beta)(2N+1)}{\ell_B^2} \end{pmatrix},$$

$$(5)$$

with $N = a^\dagger a$ and $\ell_B = \sqrt{\hbar/(eB)}$ the magnetic length.

The eigenvalues of both Hamiltonians $H_{+1}$ and $H_{-1}$ are

$$E_n^s = \gamma - \frac{2\delta|n| - s\beta}{\ell_B^2} + \text{sgn}(n)\Delta_n^s \quad (6)$$

with

$$\Delta_n^s = \sqrt{\frac{2\alpha^2|n|}{\ell_B^2} + \left(\mu - \frac{2\beta|n| - s\delta}{\ell_B^2}\right)^2}, \quad (7)$$

for Landau level (LL) index $n = \pm 1, \pm 2, \pm 3, \dots$ [valence $(-)$ and conduction $(+)$] , and

$$E_0^s = \gamma - s\mu - \frac{\delta - s\beta}{\ell_B^2} \,, \tag{8}$$

for the edge states $n = 0$ [34, 46, 47]. The associated eigenvectors are spinors containing Fock states $||n|\rangle$, that is,

$$|\boldsymbol{n}\rangle_s = \begin{pmatrix} A_n^s \left||n| - \frac{s+1}{2}\right\rangle \\ B_n^s \left||n| + \frac{s-1}{2}\right\rangle \end{pmatrix}, \tag{9}$$

with coefficients

$$A_n^s = \begin{cases} \frac{\operatorname{sgn}(n)}{\sqrt{2}} \sqrt{1 + \operatorname{sgn}(n)\cos\theta_n^s}, & n \neq 0, \\ (1 - s)/2, & n = 0, \end{cases} \tag{10}$$

$$B_n^s = \begin{cases} \frac{s}{\sqrt{2}} \sqrt{1 - \operatorname{sgn}(n)\cos\theta_n^s}, & n \neq 0, \\ (1 + s)/2, & n = 0, \end{cases}$$

where

$$\theta_n^s = \arctan\left( \frac{\sqrt{2|n|}\,\alpha/\ell_B}{\mu - \frac{2\beta|n| - s\delta}{\ell_B^2}} \right). \tag{11}$$

Depending on $\operatorname{sgn}(n)$, the coefficients $A_n^s$ and $B_n^s$ can be written as $\cos(\theta_n^s/2)$ and $\sin(\theta_n^s/2)$ [48].

The two zero Landau levels $E_0^{+1}$ and $E_0^{-1}$ belong to different Hamiltonians, that is to spin $s = +1$ and $s = -1$ respectively. The level cross condition

$$E_0^{+1} = E_0^{-1} \Rightarrow B_{\text{inv}} = \frac{\mu}{e\beta/\hbar}, \tag{12}$$

gives the critical magnetic field $B_{\text{inv}}$ which separates the quantum spin Hall and quantum Hall regimes [47]. For instance, for a QW thickness $\lambda = 7.0$ nm (see Table II), one obtains $B_{\text{inv}} \simeq 9.60$ T. This band inversion is also graphically represented in Figure 1.

It is convenient to perform a linear fit of the parameters in Table II, in order to analyze the HgTe QWs spectrum and properties for a continuous range of the thicknesses $\lambda$,

$$\begin{aligned} \mu(\lambda) &= 77.31 - 12.53\lambda, \\ \alpha(\lambda) &= 467.49 - 14.65\lambda, \\ \beta(\lambda) &= 283.58 - 138.16\lambda, \\ \delta(\lambda) &= 458.46 - 138.25\lambda, \end{aligned} \tag{13}$$

where we use the Table II units and $\lambda$ is in nanometers. The coefficient of determination is $R^2 > 0.99$ in all cases. Using $\mu(\lambda)$ in (13), we can estimate the critical HgTe thickness where the TPT occurs in the absence of magnetic field, according to the criteria in eq.(4),

$$\mu = 0 \Rightarrow \lambda_c = 6.17 \text{ nm.} \tag{14}$$

In addition, the linear fit (13) let us plot the low energy spectra (6,8) as a function of the HgTe layer thickness $\lambda$. Namely, in Figure 1, we extrapolate the linear fit (13) to the interval $[4\,\text{nm}, 8\,\text{nm}]$. The band inversion formula (12) together with the linear fit (13) yield the relation

$$\lambda_{\text{inv}}(B) = \frac{368.31 - 2.05B}{59.7 - B} \tag{15}$$

between the applied magnetic field $B$ (in Tesla) and the HgTe layer thickness $\lambda_{\text{inv}}(B)$ (in nanometers) at which the band inversion $E_0^{+1} = E_0^{-1}$ takes place. Note that $\lambda_{\text{inv}}(B) \simeq \lambda_c = 6.17$ nm for low $B \ll 1$ T, and that $E_0^{+1} = E_0^{-1} \simeq 0$ meV at this point as shows Figure 1. The thickness $\lambda_{\text{inv}}(B)$, where the band inversion happens, is a deviation of the critical thickness $\lambda_c$ in eq.(14) for $B > 0$, so it is a way to characterize TPTs when the external magnetic field is non-null.

Higher Landau levels with $|n| > 0$ have a structural change when the spinor components in eq.(10) have equal module $|A_n^s| = |B_n^s|$, that is, when the angle (11) is $\theta_n^s = \pi/2$, which implies $\mu = (2\beta|n| - s\delta)/\ell_B^2$. The valence and conduction band contributions interchange their roles at this point, hence this is a way to define a band inversion for higher Landau levels, or equivalently, we can introduce the concept higher Landau level topological phase transition (HTPT, see [48] for more details). The condition $\theta_n^s = \pi/2$ fixes a relationship between the layer thickness and the magnetic field as it happens in eq.(15),

$$\lambda_{\text{HTPT}}(B, n, s) = \frac{77.31 - 0.86B|n| + 0.7Bs}{12.53 - 0.42B|n| + 0.21Bs} \quad n \neq 0\,. \tag{16}$$

This layer thickness is higher than $\lambda_{\text{inv}}(B)$ in eq.(15) for all $B$, $n$ and $s$, and for low magnetic fields tends to $\lambda_{\text{HTPT}}(B << 1, n, s) \simeq \lambda_c$. In Ref.[48] it has been shown that quantum fluctuations and entanglement in higher Landau levels grow at the layer thickness $\lambda_{\text{HTPT}}$. The scope of the next section is to use the periodicities in the wave packet evolution as TPT and HTPT markers, and compare the critical thicknesses of this method with the ones in eq.(16).

## III. CLASSICAL AND REVIVAL TIMES IN THE TOPOLOGICAL PHASE TRANSITIONS DETECTION

The time evolution of a wave packet for the time-independent Hamiltonian of the HgTe QW (see eq.(5)) is given by

$$|\Psi(t)\rangle_s = \sum_{n=-\infty}^{\infty} c_n^s |\boldsymbol{n}\rangle_s e^{-iE_n^s t/\hbar}, \tag{17}$$

where $|\boldsymbol{n}\rangle_s$ are the eigenvectors in (9), $E_n^s$ the energies in (6), and $c_n^s = {}_s\langle n|\Psi(0)\rangle$ with $|\Psi(0)\rangle$ the initial wave packet. For the sake of simplicity, we shall take $s$ fixed,

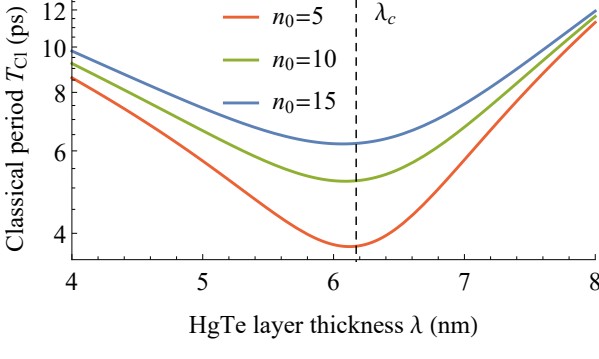

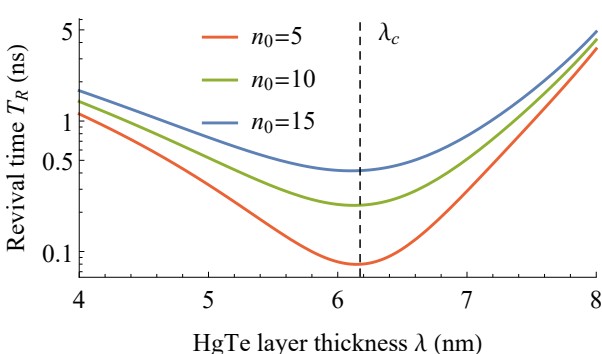

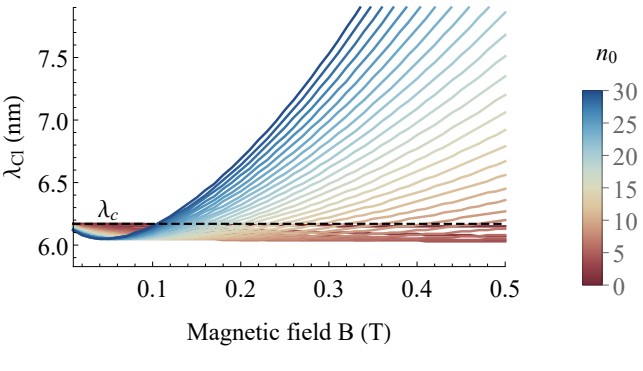

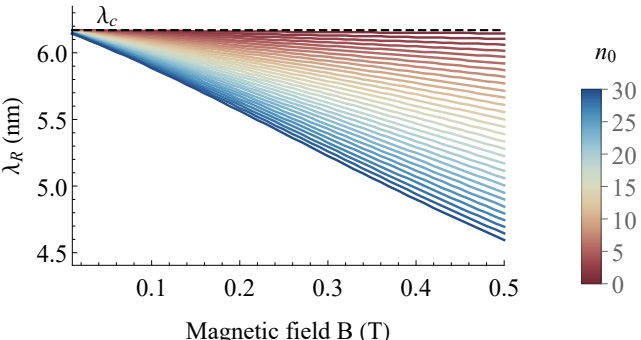

FIG. 2. Classical period $T_{\rm Cl}$ (top, eq.(20)) and revival time $T_{\rm R}$ (bottom, eq.(21)) as a function of the layer thickness $\lambda$, for three different initial wave packets $n_0 = 5, 10, 15$. In both figures, we set $B = 0.05$ T and $s = +1$, and lin-log scale. The vertical dashed line indicates the critical thickness $\lambda_{\rm c} = 6.17$ nm.

FIG. 3. Layer thicknesses $\lambda_{\rm Cl}(B, n)$ and $\lambda_{\rm R}(B, n)$ in which $T_{\rm Cl}$ and $T_{\rm R}$ achieve their minimum value respectively, as a function of the external magnetic field $B$ and for different initial wave packets $n_0 \in [0, 30]$. In both figures, we set $s = +1$ and a horizontal dashed line indicating the critical thickness $\lambda_{\rm c} = 6.17$ nm.

and $|\Psi(t)\rangle_s$, $E_n^s$, $c_n^s$ and $\lambda_{\rm HTPT}(B, n, s)$ will be referred to as $|\Psi(t)\rangle$, $E_n$, $c_n$ and $\lambda_{\rm HTPT}(B, n)$. We also select a Gaussian-like initial wave packet, distributed around a given energy $E_{n_0}$ of the spectrum $E_n$, so that

$$c_n = \frac{1}{\sigma\sqrt{2\pi}} e^{-(n-n_0)^2/2\sigma^2}, \qquad (18)$$

and we can Taylor expand the energy $E_n$ around the energy level $n_0$ [3]. Therefore, the exponential $\exp(-iE_n^s t/\hbar)$ in (17) yields

$$\exp\left(-iE_{n_0}\frac{t}{\hbar} - 2\pi i(n-n_0)\frac{t}{T_{\rm Cl}} - 2\pi i(n-n_0)^2\frac{t}{T_{\rm R}} + \dots\right) \qquad (19)$$

obtaining different time scales characterized by the classical period $T_{\rm Cl} = 2\pi\hbar/|E'_{n_0}|$ and the revival time $T_{\rm R} = 4\pi\hbar/|E''_{n_0}|$ up to second order in the series (the first term $\exp(iE_{n_0}t/\hbar)$ becomes an irrelevant global phase in eq.(17)). In fact, the classical period is the time that the wave packet needs to follow the expected semiclassical trajectory, and the revival time is the time that the wave packet needs to return approximately to its initial shape [3]. Quantum revivals are a consequence of the quantum beats [49], representing interference effects of the terms in (17). Notice that $T_{\rm Cl} << T_{\rm R}$, and thus, a

signal can be analyzed in these different regimes. Both periods have been previously studied in 2D gapped Dirac materials [4], and now we shall put the spotlight on HgTe QWs. In particular, for the energies in (6), the classical and revival periods are

$$T_{\rm Cl} = 2\pi\hbar\ell_B^2\left[-2\,{\rm sgn}(n_0)\delta + \frac{1}{\Delta_{n_0}^s}(\alpha^2 - 2\beta\chi_{n_0}^s)\right]^{-1}, \qquad (20)$$

$$T_{\rm R} = 4\pi\hbar\ell_B^4\,{\rm sgn}(n_0)\left[-\frac{4\beta^2}{\Delta_{n_0}^s} + \frac{1}{(\Delta_{n_0}^s)^3}(\alpha^2 - 2\beta\chi_{n_0}^s)^2\right]^{-1}, \qquad (21)$$

where $\chi_n^s = \mu - (2\beta|n| - s\delta)/\ell_B^2$ is the denominator in (11) and $\Delta_n^s$ is defined in eq.(7). Both periods are a function of the magnetic field $B$, the wave packet center $n_0$, the spin $s$ and the layer thickness $\lambda$ through the parameters in equation (13). The dependence on $s$ is small (mainly for low magnetic fields) and we shall set $s = +1$ in this section.

The wave packets time evolution is visualized with the autocorrelation function $A(t) = \langle\Psi(t)|\Psi(0)\rangle$, which turns into

$$A(t) = \sum_{n=-\infty}^{\infty} |c_n|^2 e^{-iE_n t/\hbar} \qquad (22)$$

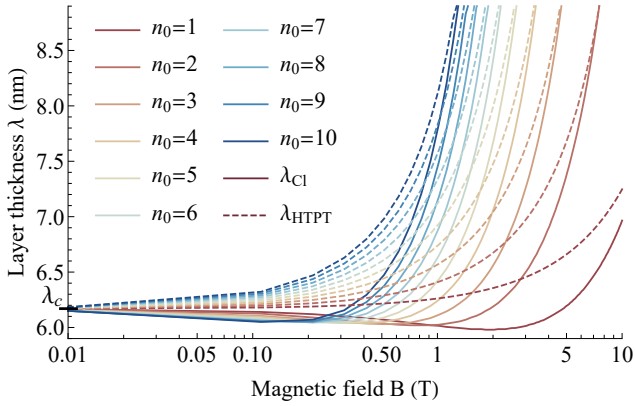

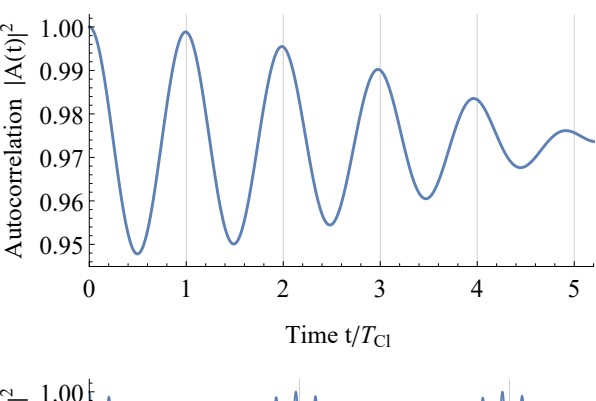

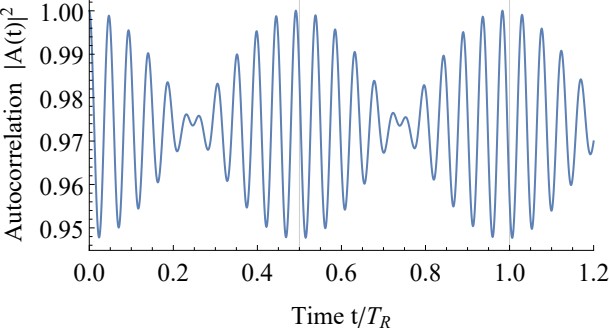

FIG. 4. Layer thickness $\lambda_{\text{Cl}}(B, n)$ in which $T_{\text{Cl}}$ achieve its minimum value (solid lines) and layer thickness of HTPT (dashed lines, see eq.(16)), as functions of the external magnetic field $B$ and for different initial wave packets $n_0 \in [1, 10]$. In both figures, we set $s = +1$ and we mark the critical thickness $\lambda_c = 6.17$ nm in the vertical axis.

for a Gaussian wave packet like the one in eq.(17).

Throughout the article, we have selected an initial wave packet localized around $n_0 = 5$ and with standard deviation $\sigma = \sqrt{n_0}/5 \simeq 0.45$ in order to analyze the wave packet evolution in (17). We have also set an external magnetic field of $B = 0.05$ T in order to observe TPT phenomena around $\lambda_{\text{inv}}(B) \simeq \lambda_c = 6.17$ nm [39]. The last restriction will be more evident later when we present Figures 3 and 9.

In Figure 2, we plot $T_{\text{Cl}}$ and $T_R$ as a function of the layer thickness $\lambda$ for spin $s = +1$ (similar results can be obtained for $s = -1$). Both periods reach a minimum near the critical thickness $\lambda_c = 6.17$ nm, hence they are useful magnitudes to identify TPTs. These minima separate from $\lambda_c$ for larger values of the magnetic field $B$ as shows Figure 3, where we present the values of the thickness $\lambda_{\text{Cl}}(B, n_0)$ and $\lambda_R(B, n_0)$ in which $T_{\text{Cl}}$ and $T_R$ achieve their minimum value respectively, as a function of the external magnetic field $B$ and the center of the wave packet $n_0$ (spin $s = +1$ fixed). For instance, for $n_0 = 5$ we obtain the bounds $|\lambda_{\text{Cl}}(B, n_0) - \lambda_c| < 0.14$ nm and $|\lambda_R(B, n_0) - \lambda_c| < 0.25$ nm in a magnetic field range $B \leq 0.5$ T. For wave packets centered in high-energy states (blue lines in Figure 3), the deviation from $\lambda_c$ is even higher, representing a criticality of the system which differs from the band-insulator phase transition. In order to characterize the criticality of $T_{\text{Cl}}$ for magnetic fields $B > 0.5$ T, in Figure 4 we compare the minimum thicknesses $\lambda_{\text{Cl}}(B, n_0)$ (solid lines) with the ones $\lambda_{HTPT}(B, n_0)$ in eq.(16) (dashed lines), where the HTPTs occurs. Both solid and dashed lines exhibit different behaviors when varying the magnetic field $B$ and the wave packet center $n_0$. Therefore, it seems that there is no correlation between the minimum thicknesses $\lambda_{\text{Cl}}$ and the HTPT at $\lambda_{HTPT}$. Nevertheless, when the magnetic field approximates to zero, both

FIG. 5. Autocorrelation function amplitude $|A(t)|^2$ as a function of time in $T_{\text{Cl}} = 3.75$ ps (top) and $T_R = 79.80$ ps (bottom) units, for an initial wave packet with $n_0 = 5$ and $\sigma = \sqrt{n_0}/5 = 0.47$. We set the HgTe parameters $\lambda = \lambda_c$, $B = 0.05$ T, and $s = +1$.

thicknesses tend to $\lambda_c = 6.17$ nm at zero field, that is, $\lambda_{\text{HTPT}}(B, n_0) \simeq \lambda_{\text{Cl}}(B, n_0) \simeq \lambda_c$ for all $n_0 \neq 0$ and $B << 1$ T.

In Figure 5, we present the squared modulus of the autocorrelation function for $\lambda = \lambda_c$ and $s = +1$ in two different time scales. The top panel displays the time in units of the classical period $T_{\text{Cl}} = 3.75$ ps, where each oscillation corresponds to one unit of the scale; whereas the bottom panel shows the wave packet revivals at half of the revival time $T_R/2 = 39.90$ ps, and the time scale is in $T_R$ units.

## IV. ELECTRON CURRENT REVIVALS AND TOPOLOGICAL PHASE TRANSITIONS

We have also identified topological phase transition by analyzing how the electron current changes with the layer thickness and the time evolution. The electron currents of the HgTe QWs have been previously studied in reference [39], where the current operators are

$$j_x^s = \frac{e}{\hbar}\left(s\alpha\tau_x - \sqrt{2}\frac{a^\dagger + a}{\ell_B}(\beta\tau_z + \delta\tau_0)\right),$$

$$j_y^s = \frac{e}{\hbar}\left(\alpha\tau_y + i\sqrt{2}\frac{a^\dagger - a}{\ell_B}(\beta\tau_z + \delta\tau_0)\right), \quad (23)$$

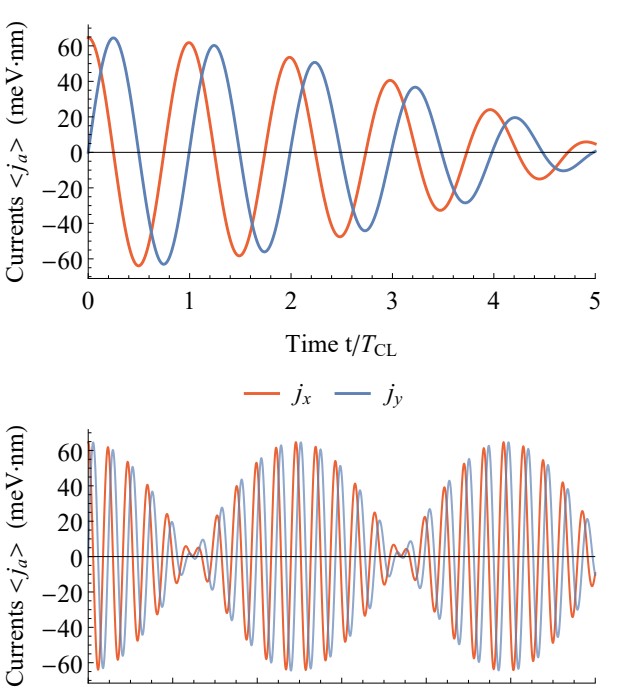

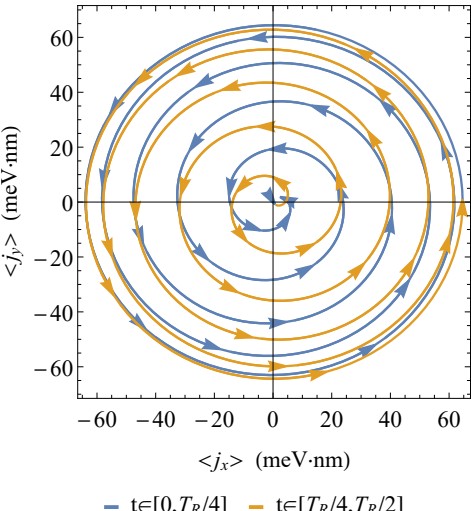

FIG. 7. Parametric plot of the currents expected values $(\langle j_x \rangle, \langle j_y \rangle)$ in the time intervals $t \in [0, T_{\mathrm{R}}/4]$ (blue) and $t \in [T_{\mathrm{R}}/4, T_{\mathrm{R}}/2]$ (yellow), for an initial wave packet with $n_0 = 5$ and $\sigma = \sqrt{n_0}/5 = 0.47$. We set the HgTe parameters $\lambda = \lambda_{\mathrm{c}}$, $B = 0.05$ T, and $s = +1$, so that the revival time is $T_{\mathrm{R}} = 79.80$ ps.

FIG. 6. Currents expected values $\langle j_a \rangle$ as a function of time in $T_{\mathrm{Cl}} = 3.75$ ps (top) and $T_{\mathrm{R}} = 79.80$ ps (bottom) units, for an initial wave packet with $n_0 = 5$ and $\sigma = \sqrt{n_0}/5 = 0.47$. The red (blue) line correspond to the current in the $x$ ($y$) axis. We set the HgTe parameters $\lambda = \lambda_{\mathrm{c}}$, $B = 0.05$ T, and $s = +1$.

and the matrix elements in the eigenstate basis (9) are

$$\langle \boldsymbol{m} | j_x^s | \boldsymbol{n} \rangle_s = \frac{es\alpha}{\hbar} \Xi_{m,n}^{s,+} - \frac{\sqrt{2}e}{\hbar \ell_B} \Phi_{m,n}^{s,+},$$

$$\langle \boldsymbol{m} | j_y^s | \boldsymbol{n} \rangle_s = -\mathrm{i}\frac{e\alpha}{\hbar} \Xi_{m,n}^{s,-} + \mathrm{i}\frac{\sqrt{2}e}{\hbar \ell_B} \Phi_{m,n}^{s,-}, \qquad (24)$$

where

$$\Xi_{m,n}^{s,\pm} = \left( A_m^s B_n^s \delta_{|m|-s,|n|} \pm A_n^s B_m^s \delta_{|m|+s,|n|} \right), \qquad (25)$$

$$\Phi_{m,n}^{s,\pm} = \left( (\delta + \beta) A_m^s A_n^s + (\delta - \beta) B_m^s B_n^s \right)$$
$$\times \left( \sqrt{|n| + 1 + \frac{s-1}{2}} \, \delta_{|m|-1,|n|} \pm \sqrt{|n| - \frac{s+1}{2}} \, \delta_{|m|+1,|n|} \right).$$

For a Gaussian wave packet (17), the electron current expected value is

$$_s\langle \Psi(t) | j_a^s | \Psi(t) \rangle_s = \sum_{m,n=-\infty}^{\infty} \overline{c_m^s} c_n^s e^{-\mathrm{i}(E_n^s - E_m^s)t/\hbar} \langle \boldsymbol{m} | j_a^s | \boldsymbol{n} \rangle, \qquad (26)$$

where $a = x, y$ and the bar indicates complex conjugation. From now on we identify $j_a^s \equiv j_a$ and choose $s = +1$ for simplicity. We plot both currents in Figure 6, for the same values $\lambda = \lambda_{\mathrm{c}}$, $B = 0.05$ T, $s = +1$, $n_0 = 5$,

$\sigma = 0.47$, as in the previous section. The results are similar to the autocorrelation in Figure 5. We observe oscillations in two different time scales, the classical ones (top panel) and the revivals (bottom panel). After half of the revival time $T_{\mathrm{R}}/2$ in the bottom panel, the electron currents reach again their maximum initial values revealing the quantum revival phenomenon. This is more evident in the phase space plot of Figure 7, where both currents decrease to zero at $t = T_{\mathrm{R}}/4$, and then they grow reaching their initial value at $t = T_{\mathrm{R}}/2$. Notice that there is a phase difference of $\pi/2$ rad between the currents $\langle j_x \rangle$ and $\langle j_y \rangle$, which is also depicted in Figure 7. The behavior shown in Figure 6 is also found in graphene [4, 50], and in 2D gapped Dirac materials under magnetic fields [39], as silicene [51].

We have repeated the calculations of Figure 6 for different values of the layer thickness $\lambda$, in order to study the impact of this parameter on the electric currents. We select the maximum of the electron current amplitudes $\mathrm{Max}_{t \in [0, T_{\mathrm{R}}/2]} |\langle j_a \rangle|$ (maximum in the time domain) for different thicknesses $\lambda$, and plot them in Figure 8. Both current maxima reach their higher value at the critical thickness. Therefore, measuring the amplitudes of the electron currents is another way to characterize TPTs in HgTe QWs. For higher magnetic fields, this maximal behavior deviates from the critical thickness $\lambda_{\mathrm{c}}$.

In Figure 9, we plot the layer thickness $\lambda_{j_a}(B, n_0)$ in which the dots $\mathrm{Max}_{t \in [0, T_{\mathrm{R}}/2]} |\langle j_a \rangle|$ of Figure 8 achieve a maximum in the $\lambda$ domain, against the external magnetic field $B$ and for an initial wave packet with $n_0 = 5$. The maxima $\lambda_{j_a}$ are close to the critical thickness $\lambda_{\mathrm{c}}$ in a region of the magnetic field, i.e. $|\lambda_{j_a}(B, n_0) - \lambda_{\mathrm{c}}| < 0.1$ nm

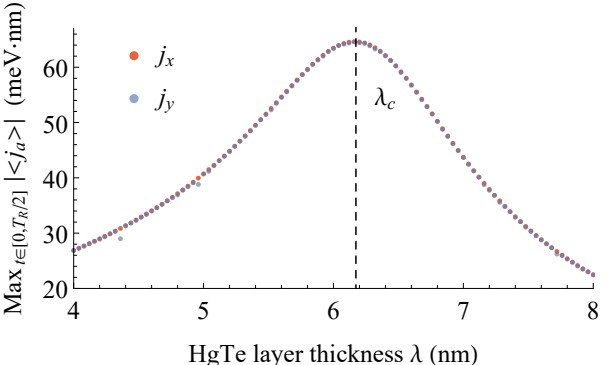

FIG. 8. Maximum values of the current amplitudes $\text{Max}_{t\in[0,T_R/2]}|\langle j_a\rangle|$ as a function of the layer thickness $\lambda$, for an initial wave packet with $n_0 = 5$ and $\sigma = \sqrt{n_0}/5 = 0.47$. The red (blue) dots correspond to the current in the $x$ ($y$) axis. We set the parameters $B = 0.05$ T, and $s = +1$.

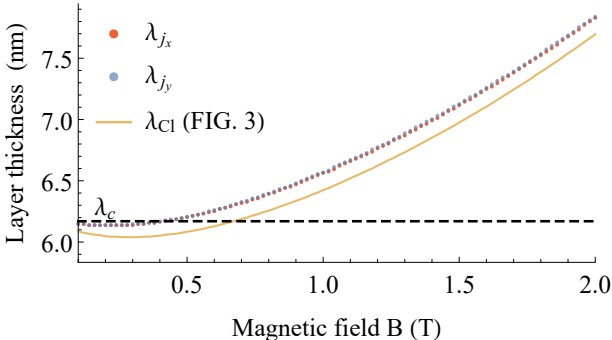

FIG. 9. Layer thickness $\lambda_{j_a}(B, n_0)$ in which the dots $\text{Max}_{t\in[0,T_R/2]}|\langle j_a\rangle|$ of Figure 8 achieve a maximum in the $\lambda$ domain, as a function of the external magnetic field $B$. The red and blue dots correspond to the directions $a = x$ and $a = y$ respectively, and the yellow line depicts the thicknesses where $T_{Cl}$ achieves its minimum (retrieved from Figure 3). We set $s = +1$ and an initial wave packet with $n_0 = 5$ and $\sigma = \sqrt{n_0}/5 = 0.47$. The horizontal dashed line indicates the critical thickness $\lambda_c = 6.17$ nm.

for all $B < 0.5$ T and $n_0 = 5$. When increasing the magnetic field above $B \simeq 0.5$ T, the maxima $\lambda_{j_a}$ (red and blue dots in Figure 9) start growing in a similar way to the thickness $\lambda_{Cl}$ where $T_{Cl}$ achieves its minimum in Figure 3.

## V. SPIN-ORBIT COUPLING EFFECTS

The model Hamiltonian (1) does not couple the spin $s = \pm 1$ blocks. It is known [52] that two different atoms in each unit cell breaks bulk inversion symmetry and leads to additional terms coupling of the spin blocks (spin-orbit interaction). In particular, terms describing the strong bulk inversion asymmetry (BIA) and

structural inversion asymmetry (SIA) have been considered in the literature (see e.g. [20, 53, 54]) and lead to novel effects. In HgTe/CdTe QWs these two types of terms are sometimes ignored because BIA terms are small when compared with the gap, and the QW are symmetric, which minimizes SIA. However, let us analyze how would a coupling term of this type affect our results. For it, let us consider the simplest spin-orbit interaction by adding to the Hamiltonian (1) a coupling BIA term proportional to $\Delta$ that leads to the four-band effective model in $2 \times 2$ block matrix form:

$$H_\Delta = \begin{pmatrix} H_{+1} & -i\Delta\tau_y \\ i\Delta\tau_y & H_{-1} \end{pmatrix}. \qquad (27)$$

The values of the spin-orbit coupling $\Delta$ for HgTe layer thickness $\lambda = 5.5, 6.1$ and $7.0$ nm are given in table II. As we did in Eq. (13) for other Hamiltonian parameters, we can perform a linear fit

$$\Delta(\lambda) = 2.52 - 0.13\lambda, \qquad (28)$$

which provides a dependence of the spin-orbit coupling $\Delta$ on the HgTe layer thickness $\lambda$. The diagonalization of the Hamiltonian $H_\Delta$ yields four eigenvalues $\mathcal{E}_n^{(\ell)}, \ell = 1, 2$, where we keep calling $n > 0$ conduction and $n < 0$ valence bands. We are ordering energies as

$$\mathcal{E}_{|n|}^{(2)} > \mathcal{E}_{|n|}^{(1)} > \mathcal{E}_{-|n|}^{(2)} > \mathcal{E}_{-|n|}^{(1)}. \qquad (29)$$

so that they tent to the energies $E_n^s$ of the uncoupled case (6) as

$$\mathcal{E}_n^{(2)} \xrightarrow{\Delta\to 0} E_n^{-1}, \ \mathcal{E}_n^{(1)} \xrightarrow{\Delta\to 0} E_n^{+1}, \qquad (30)$$

in a neighborhood of the critical thickness $\lambda_c$. In figure 10 we give the relative difference $|T-T^\Delta|/T$ for the classical and revival times with $(T^\Delta)$ and without $(T)$ spin-orbit coupling $\Delta$. In particular we are comparing

$$T_{Cl}^\Delta = \frac{2\pi\hbar}{|\mathcal{E}_{n_0}^{(2)'}|} \quad \text{with} \quad T_{Cl} = \frac{2\pi\hbar}{|E_{n_0}^{-1'}|} \qquad (31)$$

and

$$T_R^\Delta = \frac{4\pi\hbar}{|\mathcal{E}_{n_0}^{(2)''}|} \quad \text{with} \quad T_R = \frac{4\pi\hbar}{|E_{n_0}^{-1''}|}. \qquad (32)$$

We see that the spin-orbit coupling $\Delta$ has hardly any effect on classical and revival times near the critical thickness $\lambda_c$. Revival times are more sensitive than classical times to spin-orbit coupling away from critical point. Both, classical and revival, times remain minimal at $\lambda_c$ in the presence of coupling. For the considered BIA constant interaction term, we do not expect any major differences between the coupled and uncoupled case in the analysis of electron currents revivals either, although the spin-orbit coupling introduces spin mixing and one should adopt a different scheme than the one followed in section IV where we have treated both spins separately.

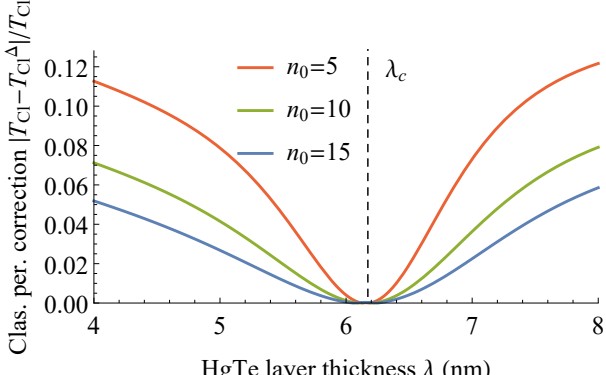

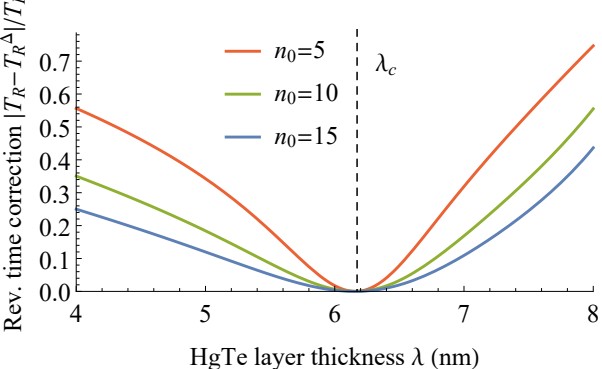

FIG. 10. Relative difference of the classical/revival times (top/bottom figure) with $(T_{\mathrm{Cl/R}}^{\Delta})$ and without $(T_{\mathrm{Cl/R}})$ spin-orbit coupling $\Delta$. Both plots are presented as a function of the layer thickness $\lambda$, for three different initial wave packets $n_0 = 5, 10, 15$. In both figures, we set $B = 0.05$ T, and $s = -1$ for the case without spin coupling. The vertical dashed line indicates the critical thickness $\lambda_{\mathrm{c}} = 6.17$ nm.

## VI. CONCLUSIONS

In summary, we have shown that the time evolution of a wave packet is useful to detect TPTs in HgTe QWs, which corroborates the results previously found in [7] for other 2D materials (silicene, germanene, tinene and indinene). Using the 2D effective Dirac Hamiltonian for surface states in HgTe/CdTe QWs, it is possible to analyze the time evolution of electron current wave packets. As a general result, the classical and revival time appear as two different periodicities in this temporal evolution, and reach their minima at different values of the layer thickness, depending on the external magnetic field and the Landau level where the packet is centered at. In addition, we have investigated how the maximum of the electron current amplitude changes with the thickness $\lambda$, identifying that current maxima reach their higher value at the critical thickness, so we can characterize the TPTs in terms of the amplitude of the electron currents. The effect of spin-orbit coupling has been addressed in section V. For small magnetic fields, we have seen that spin-orbit coupling has a negligible effect on the classical and revival times in the vicinity of the topological transition point $\lambda_c$. Both classical and revival times take minimal values at $\lambda_c$ as for the uncoupled case.

As a proposal for future work, this quantum revival analysis could be extended to non-topological anisotropic materials like phosphorene, which also present criticality when its energy gap is closed by an external electric field [39].

## ACKNOWLEDGMENTS

We thank the support of Spanish MICIU through the project PID2022-138144NB-I00. AM thanks the Spanish MIU for the FPU19/06376 predoctoral fellowship. OC is on sabbatical leave at Granada University, Spain, since the 1st of September 2023. OC thanks support from the program PASPA from DGAPA-UNAM.

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
