# Peer review of "Quantum revivals in HgTe/CdTe quantum wells and topological phase transitions"

_SciPost Physics Core_

## Round 1 · Referee Report · Anonymous (Referee 1) · 2024-4-1

Strengths

Innovative approach to detect topological phase transitions, applied to a material of experimental relevance.

Weaknesses

The model Hamiltonian is somewhat oversimplified.

Report

This work is a follow-up of Ref. 7, where the authors develop the general theory. Here the focus is on an application to a particular material, which is interesting and worth publishing, but lacks the "breakthrough" character required for SciPost Physics. It seems more appropriate for SciPost Physics Core.

One issue would be worth considering: the model Hamiltonian which the authors study does not couple the spin blocks, see Eq. 1. It is known that spin-orbit interaction without inversion symmetry introduces a coupling of the spin blocks. Even if the quantum well is symmetric, there remains a bulk inversion asymmetry. How would a coupling term affect the results?

Requested changes

Assess the effect of coupling of the spin blocks, at least qualitatively, if possible, quantitatively.

  • validity: good
  • significance: good
  • originality: good
  • clarity: good
  • formatting: good
  • grammar: good

Author:  Alberto Mayorgas  on 2024-04-19  [id 4434]

(in reply to Report 1 on 2024-04-01)
Category:
correction

Dear Referees,

Following the referee suggestions, we have added a new section V to study the effect of spin-orbit interaction without inversion symmetry introducing a constant coupling of the spin blocks. We focused on the effect of spin-orbit interaction on classical and revival times. We have seen that the spin coupling has hardly any effect on classical and revival times near the critical thickness where the topological phase transition takes place. Revival times turn to be more sensitive than classical times to the coupling away from the critical point. Both, classical and revival, times remain minimal at the critical point. More details (with new figures) can be found in the new version of the manuscript.

We hope that the changes introduced in the manuscript will make it suitable for publication by SciPost Physics Core.

Sincerely,

The authors

---

## Round 2 · List of Changes

We have added a new section V to study the effect of spin-orbit interaction without inversion symmetry introducing a constant coupling of the spin blocks. We focused on the effect of spin-orbit interaction on classical and revival times. We have seen that the spin coupling has hardly any effect on classical and revival times near the critical thickness where the topological phase transition takes place. Revival times turn to be more sensitive than classical times to the coupling away from the critical point. Both, classical and revival, times remain minimal at the critical point. More details (with new figures) can be found in the new version of the manuscript.

---

## Editorial Decision

editorial_decision: